# Pillared Graphene Structures Supported by Vertically Aligned Carbon Nanotubes as the Potential Recognition Element for DNA Biosensors

**DOI:** 10.3390/ma13225219

**Published:** 2020-11-19

**Authors:** Vladislav V. Shunaev, Olga E. Glukhova

**Affiliations:** 1Department of Physics, Saratov State University, 410012 Saratov, Russia; shunaevvv@sgu.ru; 2Institute for Bionic Technologies and Engineering, I.M. Sechenov First Moscow State Medical University (Sechenov University), 119991 Moscow, Russia

**Keywords:** biosensor, pillared graphene, DNA, conductivity, mathematical modeling

## Abstract

The development of electrochemical biosensors is an important challenge in modern biomedicine since they allow detecting femto- and pico-molar concentrations of molecules. During this study, pillared graphene structures supported by vertically aligned carbon nanotubes (VACNT-graphene) are examined as the potential recognition element of DNA biosensors. Using mathematical modeling methods, the atomic supercells of different (VACNT-graphene) configurations and the energy profiles of its growth are found. Regarding the VACNT(12,6)-graphene doped with DNA nitrogenous bases, calculated band structure and conductivity parameters are used. The obtained results show the presence of adenine, cytosine, thymine, and guanine on the surface of VACNT(12,6)-graphene significantly changes its conductivity so the considered object could be the prospective element for DNA biosensing.

## 1. Introduction

DNA and microRNA play a key role in storing and transmitting genetic information. Currently, molecular analysis of DNA can diagnose more than 400 diseases [1]. It is known that early determination of some diseases, such as cancer, increases patient chances of survival [2,3,4]. Therefore, an essential biomedical tool is highly sensitive biosensors that can detect femto- and pico-molar concentrations of molecules. Due to their unique absorption, electrical and conductive properties, as well as high mechanical strength and flexibility, nanocarbon structures such as graphene and carbon nanotubes are important components of modern biosensors [5]. There are many types of graphene and CNT-based biosensors: for example, fluorescence [6,7,8,9], surface enhanced Raman scattering (SERS) [10,11,12,13], and surface plasmon resonance biosensors (SPR) [14,15,16,17,18]. However, electrochemical biosensors are more often used to detect biomarkers such as DNA and microRNA [19]. They represent two- or three-electrode electrochemical cells that contain a biological recognition element generating electrochemical signals. One of the first prototypes of CNT-based biosensors of specific DNA was proposed by Cai et al. [20]. Due to fast electron transfer between CNTs–COOH-modified glassy carbon electrodes (GCE) and oligonucleotide probes with an amino group at its 50-phosphate end (NH_2_-ssDNA), the detection sensitivity achieves 1.0 × 10^−10^ mol L^−1^ for complementary oligonucleotide. The application of impedance technology has reduced the detection of complementary DNA sequence limits to 5 × 10^−12^ mol L^−1^ [21]. The discovery of superconducting graphene allows for a significant step in the development of electrochemical biosensors, to increase the detection of DNA limits to 9.4 × 10^−21^ mol L^−1^ [22], and for the detection of miRNA limits to 0.2 × 10^−15^ mol L^−1^ [23].

The high sensitivity of CNTs coupled with the larger detecting surface area of graphene makes the hybrid material on their base an ideal candidate for the role of a biosensor element. Zou et al., fabricated a novel glucose biosensor by loading a glassy carbon electrode with a 1-methyl imidazole-based ionic liquid-functionalized graphene/CNT composite [24]. The biosensor had the detection limit of 3.99 × 10^−7^ mol/L and a sensitivity of 53.89 μA mmol/L^−1^cm^−2^ for glucose detection with an excellent stability and reproducibility. An electrochemical chiral sensor via an integrated polysaccharides/3D nitrogen-doped graphene-CNT frame exhibited excellent reproducibility, stability, and selectivity [25]. Mousaabadi et al., proposed reduced graphene oxide and carbon nanotube composite functionalized by azobenzene that could detect curcumin in the range of 0.008 to 10.0 μM with the detection limit of 0.003 μM [26]. Three-dimensional (3D) graphene-based wearable piezoresistive sensors were considered promising flexible sensors due to their facile preparation, simple read-out mechanism, and low power consumption [27]. Zhao et al., first synthesized an MoS_2_-CNTs@GONR nanocomposite that exhibited satisfactory stability and accuracy for quercetin detection in actual samples [28].

Methods of mathematical modeling results serve as a guideline for the design and optimization of biosensors on the base of graphene and CNTs. Khodadadi et al., fabricated an epirubicin biosensor based on the interactions of DNA with polypyrrole and nitrogen-doped reduced graphene during the study of docking theoretical investigations confirming the binding of epirubicin and guanine sides in the ds-DNA structure [29]. The analytical modeling of single-walled carbon nanotube-field effect transistors (SWCNT-FETs) biosensors for glucose detection showed a good agreement with experimental data [30]. Article Neural Network and analytical modeling have been employed to demonstrate the presence of NH_3_ gas significantly increased the conductivity of CNTs [31]. This effect can be used in gas sensor devices. The molecular dynamics method using a CHARMM27 force field allowed for the study of the interaction of variously charged SWNT with flavin adenine dinucleotide (FAD) [32]. Studies have shown that the attaching of FAD to uncharged and positively charged CNTs increased the steric resistance of molecules. Tian et al., designed a graphene field effect transistor biosensor to detect the binding interactions between guanine riboswitch (GR) and four purine analog ligand molecules (GUA, 6GU, 2BP, XAN) [33]. To reveal the sensing mechanism of the proposed sensor and the conformational changes of the GR aptamer after binding to the ligand molecules GUA, 6GU, 2BP and XAN, we calculated using molecular dynamic simulation. Density functional theory (DFT) calculations showed that adsorption of dopamine on nitrogen-Fe-doped single carbon nanotube (8,0) led to changes to the energy gap [34]. The simulation results of electrical conductivity changes showed DNA/Cu_2_O-graphene sheet nanostructures can be well used as FET to detect polycyclic aromatic hydrocarbons [35].

Recently, such hybrid 3D materials as pillared graphene structures supported by vertically aligned CNTs (VACNT-graphene) have become increasingly popular. This material is successfully used for hydrogen storing and thermal transport and is used in nanoelectronic devices and supercapacitors [36,37,38,39]. Ke Duan et al., reported that mass sensitivity of the pillared graphene structure can reach at least 1 yg (10^−24^ g), which is even better than the carbon nanotube-based nanosensors [40]. Here, mathematical modeling methods will be used to make a predictive assessment of the potential possibility for the formation of vertically oriented single-layer nanotubes (SWCNTs) of various chiralities in the holes of nanomesh graphene with a seamless bonding type. The VACNT(12,6)-graphene composite will be considered a potential candidate for the role of a DNA component molecule biosensor.

## 2. Methods and Results

### 2.1. Construction of VACNT-Graphene Atomic Supercells

To build atomic structures of VACNT-graphene, graphene nanoparticles with sizes 34.44 Å × 34.08 Å (for simplicity, we will denote these dimensions as «34 × 34») and SWCNTs with chirality indexes (12,6), (16,0), (14,4), (11,10), (12,6), (6,5) were used as the most often synthesized [41,42,43]. Regarding the construction of seamless junctions, the original method was applied for generating atomistic models of multi-branched and arbitrary-shaped seamless junctions of carbon nanostructures [44]. Consider the example of atomic supercells construction on the example of VACNT(12,6)-graphene (Figure 1). Occurring at the initial stage, to obtain graphene nanomesh (GNM) in the center of a graphene nanoparticle, a hole was cut in the form of an ellipse with the axes 2a = 18.5 Å and 2b = 19.64 Å. Concerning the area of this hole, as well as at the edge of the bonded SWCNT, atoms with only two covalent bonds—a so-called TNMF (triangulated nanomesh frame)—were distinguished. Furthermore, the coordinates of the new atoms were randomly generated between the TNMF atoms. To accelerate the formation of the contact region between the graphene and SWCNT, the modified molecular dynamics method was applied and its motion equation contained an additional term responsible for the attraction of external atoms to the TNMF surface. Subsequently, the coordinates of the resulting structure were refined using the self-consistent charge density functional tight-binding (SCC DFTB) method, which was further used to calculate the energy and electronic properties [45]. Note that for each compound, five variants of atomic mesh were constructed and only one of them (with the lowest energy) was chosen for subsequent studies. The most energy-favorable supercells of VACNT-graphene with SWCNTs (16,0), (14,4), (11,10), (12,6), (6,5) are shown in Figure 2. Only VACNT(12,6)-graphene has a pentagon defect in its structure, while other supercells contain only heptagon defects. These results confirm the suggestion that the formation of heptagons in the contact area between SWCNT and graphene is more energy favorable than the formation of pentagons [46].

Occurring at the next stage, the energy profile of the considered junction’s growth was constructed. To this aim, the author-developed technique called “virtual growing” was used, which was previously successfully tested for seamless junctions on the base of GNMs and SWCNTs of the armchair-type [47]. Regarding the top layer of carbon atoms, a step-by-step removed case from the edge of the SWCNT component of pillared graphene was used, while the atomic structure of initial GNM was not achieved. Figure 3 demonstrates the final stages of this process using the example of the VACNT(12,6)-graphene. Each step structure was reoptimized using the SCC DFTB method, and the energy ΔH = E_i_ − E_0_ was calculated, where E_0_ was the energy per atom of the initial GNM and E_i_ was the energy per atom of the current structure. Figure 4a shows the growth energy profile of the considered VACNT-graphene structures that represent the dependence of ΔH on the number of layers. Formation of all junctions is energy favorable. Upon 3–5 layers of growth, the splash of energy is observed, but then it decreases for all cases except the VACNT(6,5)-graphene.

### 2.2. Finding the Optimal Sizes of the VACNT(12,6)-Graphene Supercell

The SWCNT(12,6) is the only tube with conductive properties among those considered, so the junction on its base was considered as the potential biosensor. Probably, the electronic structure of this particular compound will react more than others to the presence of other molecules near its structure. Since surface area and intertube bridging are key parameters for different properties of pillared graphene [48,49,50], the search of the VACNT(12,6)-graphene supercell optimal sizes was performed. To this aim, the energy profiles of the growth for the VACNT(12,6)-graphene atomistic structures with dimensions 44.36 Å × 42.63 Å and 49.32 Å × 51.17 Å (further for simplicity—«44 × 44» and «50 × 50») also was calculated. Figure 4b shows the growth of the composites with these sizes is possible but less energy favorable in comparison to the case of the 34 × 34 structure. To explain the reason for this difference, we calculated the charge distribution over the atoms of the initial GNM supercells according to the Mulliken procedure, previously successfully used for calculating the charges on the atoms of other composite carbon nanostructures [51,52]. Figure 5 shows the distribution of charge over the initial GNM atoms near the hole for different sizes of supercell. Regarding a 34 × 34 GNM, the charges on atoms near the hole significantly exceed the charges on atoms near the hole in GNMs with larger dimensions. Consequently, these atoms are more chemically active, which explains the more favorable growth of VACNTs precisely from the hole in this structure.

### 2.3. Influence of DNA Nitrogenous Bases on Electric Properties of the VACNT(12,6)/Graphene

Since the DNA molecule is too small for quantum chemical calculation, we performed a study of the interaction between the VACNT(12,6)-graphene and nitrogenous bases: guanine, adenine, thymine, and cytosine often abbreviated by G, A, T and C, respectively. The search for the optimal location of nitrogenous bases on the surface of the VACNT(12,6)-graphene supercell was performed using the SCC DFTB 2 method using the 16 × 16 × 16 Monkhorst–Pack sampling and 3ob-3-1 parameterization [53]. As a result of a series of numerical experiments, two most advantageous locations of nitrogenous bases on the surface of the carbon composite with translation vectors 34 × 34 were determined. Taking into account that the landing sites for all nitrogenous bases are approximately the same, Figure 6 shows only the optimal locations of adenine on the surface of the VACNT(12,6)-graphene. Landing Site 1 corresponds to the position opposite the pentagon, formed during the composite’s construction. The Mulliken charge distribution shows the greatest shifts of electron clouds are observed particularly in the place that enhances the interaction of the carbon atoms in the region with the atoms of nitrogenous bases. Site 2 is located inside the CNT. Nitrogenous-based molecules have the ability for a van der Waals interaction with a greater number of carbon atoms at this position.

Band structure analysis showed that addition of adenine and guanine to the surface of the VACNT(12,6)-graphene decreases its Fermi level from −4.44 to −4.87 eV, while the addition of just adenine lowers to −4.88 eV, and of just thymine further reduces to −4.89 eV. Figure 7 shows the presence of nitrogenous bases significantly changes the density of states (DOS) of the composite and leads to the appearance of the noticeable peak at the Fermi level that notes the conductance growth in this area. Note that the DOS of the pure composite touched zero at the E − E_f_ = −0.4 eV, and for the composite decorated with adenine, thymine, or cytosine this value shifted to −0.24 eV. The exception was the addition of guanine—in this case the noticeable peak at −0.24 eV is observed. It also draws attention to the fact that the DOS of VACNT(12,6)-graphene+A and VACNT(12,6)-graphene+T are similar. This may be caused by the fact that adenine and thymine form a base pair in DNA [54].

### 2.4. Influence of DNA Nitrogenous Bases on Conductive Properties of the VACNT(12,6)/Graphene

The electron transmission function T(E), electric resistance R, and electrical conductivity G were calculated at 300 K on the basis of the Keldysh nonequilibrium Green function method [55] and Landauer–Büttiker formalism [56]. This method, described in our previous paper [47], allowed us to calculate the T(E) and G functions for super-cells within 1000 atoms. The dependence of T(E) on energy for pure and decorated VACNT(12,6)-graphene structures is shown in Figure 8 (the values of T(E) are given in the conductance quantum e^2^/h). Similar to the case of seamless junctions between SWCNT and graphene of the armchair-type, the anisotropy of transmission is not observed [47]. When the current flew along the zigzag directions, the plateau at the value T(E) = 1 was observed near Fermi level (Figure 8a–d) for all cases. That caused the increase in conductivity G for composites with nitrogenous bases by 5–10%, in comparison to the clean sample (Table 1). When the current flew along the armchair directions, the situation changed. T(E) at the Fermi level decreased to ~0.3 (Figure 8e–h) and the conductivity G decreased by 38–45% (Table 1).

To explain the presence of peaks on the graph of the integral characteristic T(E) (Figure 8), we calculated the transmission function T(k,E) maps (Figure 8); in this case, the summation was performed along all the wave numbers k. The wave number k corresponded to the direction along which electronic transport was conducted. Figure 9 clearly shows, regardless of the nitrogenous base type, the value of T(k,E) at the Fermi level in the zigzag direction did not exceed 0.6. The only exception was observed in the case of VACNT(12,6)-graphene+G where T(k,E) is about 1 when 0.05 < k < 0.09. Along the zigzag direction, T(k,E) at the Fermi level exceeds 1 for almost any k that leads to the appearance of the plateaus in the integral characteristics T(E). We also note that the maximum values of T(k,E) along the zig-zag direction exceeding 1.4 are observed to the right of the Fermi level and at k < 0.05, and along the armchair direction—at k > 0.07.

## 3. Conclusions and Discussion

The atomic supercells of VACNT-graphene with SWCNTs of the most often synthesized chirality indexes (16,0), (14,4), (11,10), (12,6), (6,5) were obtained using the original molecular modeling methods. Our calculations showed that the growth of all considered structures was energetically favorable, which opens a prospect for its potential manufacturing. Containing semiconductive tubes, the VACNT(12,6)-graphene structure is a prospective candidate for the role of a biosensor element. Its geometrical structure facilitates adsorption of different molecules on its surface. The first place for such absorption is located in the area of defects formed at the graphene and CNT junction during growth. Another place was found in the pores of CNTs. The presence of conductive CNT(12,6) and superconductive graphene allows the assumption that conductive properties of the composite can be changed by adsorption of different molecules. *Ab initio* calculations confirmed that band structures, as well as the conductive properties of this junction, were sensitive to the presence of the DNA nitrogenous bases near its surface. Absorption of guanine, adenine, cytosine, and thymine led to the appearance of the noticeable peak at the Fermi level where they significantly affected the composite’s conductivity. When the current flew along the zigzag directions, the presence of the DNA nitrogenous bases increased the conductivity of the VACNT(12,6)-graphene by 5–10%, while, when the current flew along in the armchair direction, its conductivity dramatically decreased by 38–45%. The obtained results indicate that the VACNT(12,6)-graphene composite may be a prospective candidate for the role of a recognition element in electrochemical biosensor of DNA. The future task is to provide studies of interaction between VACNT-graphene and large DNA molecules. Since the typical DNA molecules are large, the sizes of VACNT-graphene should be enlarged too. Our preliminary calculations, for example, showed adsorption of B-DNA dodecamer [57] on the surface of the VACNT-graphene led to a giant charge transfer of about 7e that inevitably made dramatic changes in composite’s conductance. Such shifts in the composite’s conductance can be easily recognized using an external device. Naturally, during synthesis, the VACNT(12,6)-graphene composite may contain some interstitial or substitutional metal impurities, like graphene. These defects can affect the adsorption, electronic and conductive properties of the composite. The influence of different impurities on biosensing of VACNT-graphene also may be a prospective and important theme of study.

## Figures and Tables

**Figure 1 materials-13-05219-f001:**
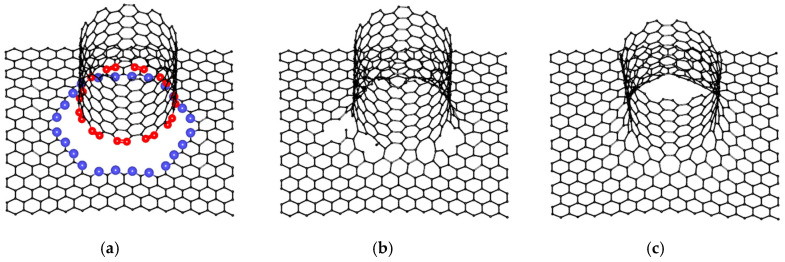
The process of VACNT(11,10)-graphene supercell construction: (**a**) determination of TNMF atoms at the edge of CNT and in the area of GNM’s hole; (**b**) generation of atoms in the space between CNT and GNM; (**c**) the final form of the supercell after SCC DFB optimization.

**Figure 2 materials-13-05219-f002:**
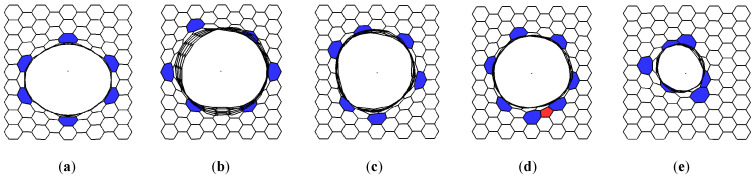
The atomic supercells of VACNT-graphene with the following indexes of CNT’s chirality: (**a**) (16,0), (**b**) (14,4), (**c**) (11,10), (**d**) (12,6), (**e**) (6,5)—the view from above. The heptagon defects are highlighted in blue and pentagons in red.

**Figure 3 materials-13-05219-f003:**
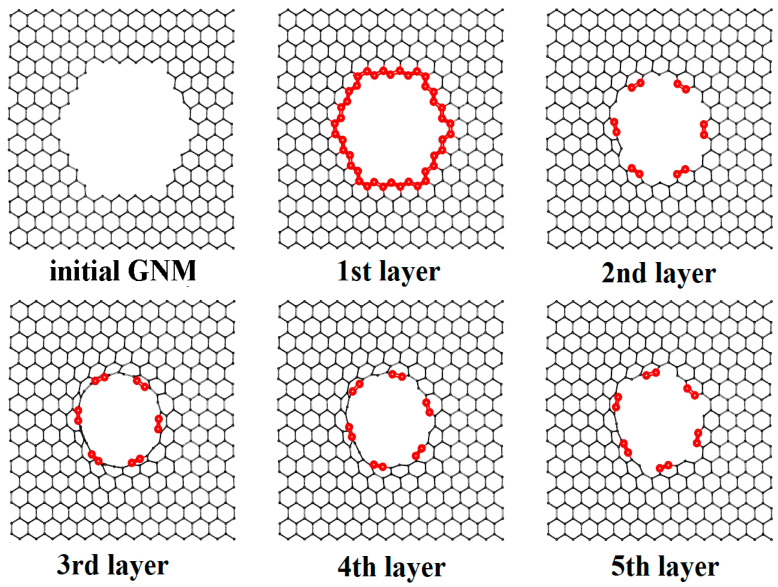
«Virtual growing» of atomistic structure VACNT(12,0)/graphene.

**Figure 4 materials-13-05219-f004:**
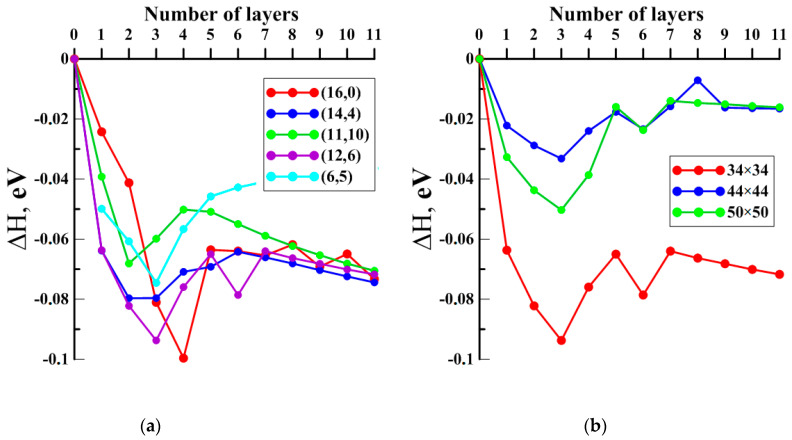
Energy profile of the growth (**a**) for VACNTs/graphene with indexes of CNT’s chirality: (16,0), (14,4), (11,10), (12,6), (6,5); (**b**) for VACNTs(12,6)/graphene with different lattice vectors.

**Figure 5 materials-13-05219-f005:**
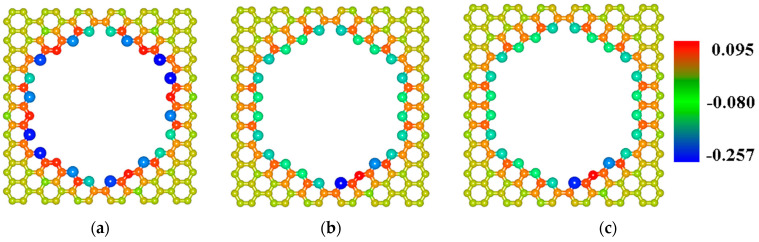
Charge distribution near GNM with lattice vectors: (**a**) 34.44 Å × 34.08 Å; (**b**) 44.36 Å × 42.63 Å; (**c**) 49.32 Å × 51.17 Å.

**Figure 6 materials-13-05219-f006:**
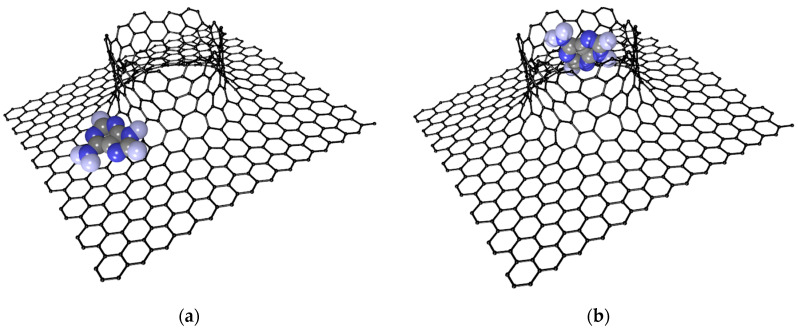
The most energy favorable locations of the adenine molecule on the VACNT(12,6)-graphene surface: (**a**)—Site 1, (**b**)—Site 2.

**Figure 7 materials-13-05219-f007:**
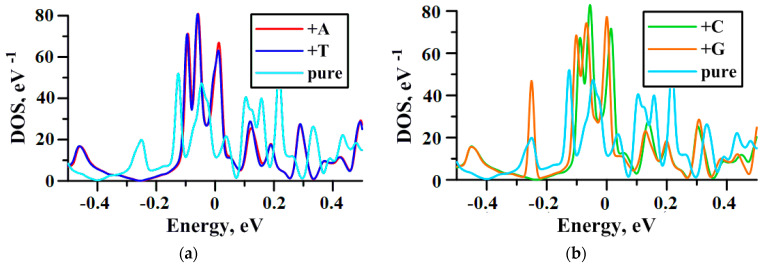
The DOS of pure VACNT(12,6)/graphene and decorated with nitrogenous bases: (**a**) with adenine and thymine; (**b**) with cytosine and guanine.

**Figure 8 materials-13-05219-f008:**
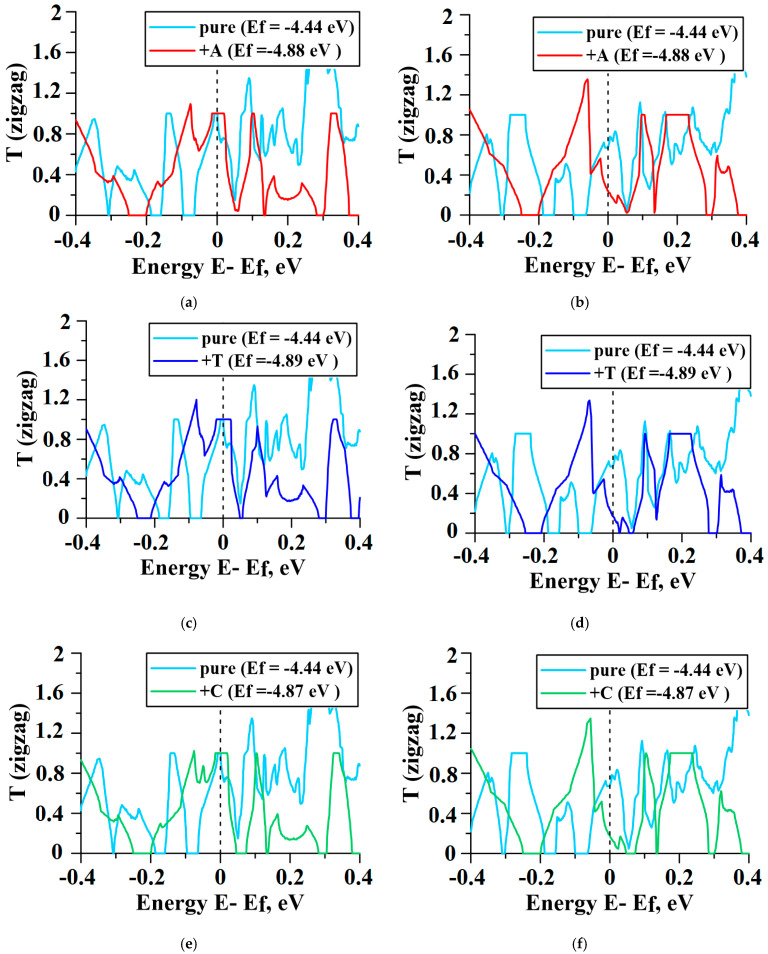
The DOS of VACNT(12,6)/graphene—pure and decorated with nitrogenous bases: (**a**,**b**) adenine; (**c**,**d**) thymine; (**e**,**f**) cytosine; (**g**,**h**) guanine.

**Figure 9 materials-13-05219-f009:**
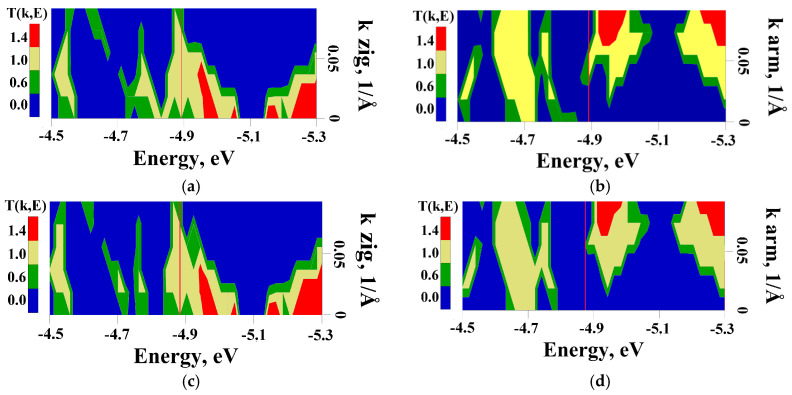
The T(k,E) map in the coordinates of the wave number k and energy E in zigzag and armchair directions for VACNT(12,6)/graphene + (**a**,**b**) adenine; (**c**,**d**) cytosine; (**e**,**f**) guanine; (**g**,**h**) thymine. Red vertical line corresponds to the Fermi level *E_f_*.

**Table 1 materials-13-05219-t001:** The conductance and resistance values of VACNT(12,6)/graphene—pure and decorated with nitrogenous bases—in zigzag and armchair directions.

	G (zig), S	R (zig), kOhm	G (arm), S	R (arm), kOhm
Pure Carcass	5.3594 × 10^−5^	18.658	4.39427 × 10^−5^	22.756
Cytosine	5.6348 × 10^−5^	17.746	2.71896 × 10^−5^	36.778
Thymine	5.9794 × 10^−5^	16.724	2.53869 × 10^−5^	39.390
Guanine	5.8864 × 10^−5^	16.988	2.7131 × 10^−5^	36.858
Adenine	5.8364 × 10^−5^	17.133	2.4219 × 10^−5^	36.739

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
