# Peer review of "Pillared Graphene Structures Supported by Vertically Aligned Carbon Nanotubes as the Potential Recognition Element for DNA Biosensors"

_materials, 2020, doi:10.3390/ma13225219_

Round 1

Reviewer 1 Report

Dear Editor,

I carefully revised the paper "Pillared Graphene Structures Supported by Vertically Aligned Carbon Nanotubes as the Potential Recognition Element for DNA Biosensors " by Shunaev and co-workers. They propose this new material to be used in DNA biosensors; they studied different possible assembles and the conditions that made them suitable for the recognition of nitrogenous bases by conductivity modulation.

The simulations and models are definitely fine, and the amount of information reported seems complete.

I would like to suggest the acceptance of the paper, after the following changes are applied:

1) I would make the fact that the paper only reports simulations more explicit in the title and the abstract of the paper;

2) I would like to see a comment about how the proposed material could be specific to a certain DNA sequence. Indeed, being sensitive to a specific analyte is not sufficient to define a material suitable for biosensing. Do authors think that a probe sequence is needed? If so, physical absorption or a kind of functionalization could be preferable? Can they foresee how this would change the situation they are now proposing?

3) authors must solve several typos in the text: here some examples (please go through a thorough reading):

  • line 32: NH2 must have '2' subscipted;
  • line 46: graphene/(CNTs; is the bracket a typo?
  • line 95: what's that symbol at the end of the line?
  • line 194: watch out for the Table 1 caption.
  • line 226: please insert author contribution.

Author Response

Dear Reviewer!

We thank you for your kind comments and advices for improving our manuscript. Here are answers on your questions and comments.

  • I would make the fact that the paper only reports simulations more explicit in the title and the abstract of the paper;

With all due respect to Reviewer, we don’t think that the title of the paper should be changed since it fully reflects the content of the manuscript. Another Reviewers also don’t suggest to change the title. Bu we mentioned about simulation character in the text. The Abstract now contains the phrase «Within mathematical modeling methods» (lines 13-14), the Conclusion contains the phrase «Ab initio calculations confirmed» (line 221)

  • I would like to see a comment about how the proposed material could be specific to a certain DNA sequence. Indeed, being sensitive to a specific analyte is not sufficient to define a material suitable for biosensing. Do authors think that a probe sequence is needed? If so, physical absorption or a kind of functionalization could be preferable? Can they foresee how this would change the situation they are now proposing?

To answer this question we performed additional calculation of interaction between pillared graphene and B-DNA molecule consisted of nucleotide base. This molecule was chosen since there are a lot of studies that confirmed its physical adsorption on graphene structure and one of such papers was added to References (Ref. 57). The large electron transfer (about 7e) between the VACNT-graphene and B-DNA was observed that indicated on shifts in object conductance. The following text was added to Conclusions (line 227-235):

The obtained results indicate that the VACNT(12,6)-graphene composite may be a perspective candidate on the role of recognition element in electrochemical biosensor of DNA. The future task is to perform calculations of interaction between VACNT-graphene and real DNA sequence molecule. Since the typical DNA molecules are big, the sizes of VACNT-graphene should be enlarged too. Our preliminary calculations showed that adsorption of B-DNA dodecamer [57] on the surface of the VACNT-graphene lead to giant charge transfer about 7e that inevitably made dramatic changes in composite’s conductance. Such shifts in composite’s conductance can be easily recognized by external device.

  • Authors must solve several typos in the text: here some examples (please go through a thorough reading):
  • line 32: NH2 must have '2' subscipted;
  • line 46: graphene/(CNTs; is the bracket a typo?
  • line 95: what's that symbol at the end of the line?
  • line 194: watch out for the Table 1 caption.
  • line 226: please insert author contribution.

All typos were repaired and highlighted by yellow. Thank you.

Reviewer 2 Report

The authors reported that pillared graphene structures supported by vertically aligned carbon nanotubes (VACNT) as the potential recognition element of DNA biosensors. Mathematical modelling showed that the atomic supercells of different vertically aligned carbon nanotubes configurations and energy profiles its growth. They have calculated the band structure and conductivity parameters for VACNT(12,6)-graphene doped with DNA nitrogenous bases and found that the presence of adenine, cytosine, thymine, and guanine on the surface of VACNT(12,6)-graphene significantly changed its conductivity demonstrating that the VACNT(12,6)-graphene composite can be considered as the perspective element in the electrochemical biosensor of DNA. This is an interesting work and can be accepted for publication. Minor comments.

-It is better if the authors could highlight the importance of vertically aligned CNT in this study.

-Graphene usually contains impurities in a true experimental sense. Any comments on the role of such intrinsic impurities in the biosensing would be nice.

-I did not see panels (a), (b), and (c) in Figure 1.

-Figure 4a should revise. Some data points are not visible.

Author Response

Dear Reviewer!

We thank you for your kind comments and advices for improving our manuscript. Here are answers on your questions and comments.

  • It is better if the authors could highlight the importance of vertically aligned CNT in this study;

 The following text was added to Conclusion (line 215-221):

Containing semiconductive tube the VACNT(12,6)-graphene structure is a prospect candidate on the role of biosensor element. Its geometrical structure facilitates adsorption of different molecules on its surface. The first places for such absorption locate in the area of defects formed in the place of graphene and CNT junction during growing. Another places were found in the pores of CNTs. The presence of conductive CNT(12,6) and superconductive graphene assumes that conductive properties of the composite can be changed by adsorption of different molecules. 

  • Graphene usually contains impurities in a true experimental sense. Any comments on the role of such intrinsic impurities in the biosensing would be nice.

The influence of impurities on biosensing properties of the VACNT-graphene was not considered in this paper though it could be the interstinf theme for researchin future. The following text was added to Conclusion (line 235-238)

Of course, during synthesis VACNT(12,6)-graphene composite may contain some interstitial or substitutional metal impurities, like graphene. These defects can affect adsorption, electronic and conductive properties of the composite. The influence of different impurities on biosensing of VACNT-graphene may be also prospect and important theme of study. 

  • I did not see panels (a), (b), and (c) in Figure 1.

Corresponding panels were inserted to Figure 1.

  • Figure 4a should revise. Some data points are not visible.

 Figure 4a was revised

Reviewer 3 Report

The present paper is interesting and it is worth of publication.

Anyway some comments and revision are necessary:

-it is not clear how this material can be used for the development of an electrochemical biosensor. I don't understand if the teoretical interaction with DNA bases was performed to prove that this material can be used to "capture" target DNA molecules in the sample. Or otherwise this graphene structure can be use to develop an innovative platform to develop an assay or genosensor. Please explain better in the text.

-The oveall manuscript needs a revision fro some typing errors. Please amend.

Author Response

Dear Reviewer!

We thank you for your kind comments and advices for improving our manuscript. Here are answers on your questions and comments.

  • it is not clear how this material can be used for the development of an electrochemical biosensor. I don't understand if the teoretical interaction with DNA bases was performed to prove that this material can be used to "capture" target DNA molecules in the sample. Or otherwise this graphene structure can be use to develop an innovative platform to develop an assay or genosensor. Please explain better in the text.

The following explanation was added to Conclusions (line 227-235):

The obtained results indicate that the VACNT(12,6)-graphene composite may be a perspective candidate on the role of recognition element in electrochemical biosensor of DNA. The future task is to provide studies of interaction between VACNT-graphene and large DNA molecule. Since the typical DNA molecules are large, the sizes of VACNT-graphene should be enlarged too. For example, our preliminary calculations showed that adsorption of B-DNA dodecamer [57] on the surface of the VACNT-graphene lead to giant charge transfer about 7e that inevitably made dramatic changes in composite’s conductance. Such shifts in composite’s conductance can be easily recognized by external device.

  • The oveall manuscript needs a revision fro some typing errors. Please amend.

The text of the paper was carefully checked; some typing errors were repaired and highlighted by yellow.
